# The Genus *Capsicum*: A Review of Bioactive Properties of Its Polyphenolic and Capsaicinoid Composition

**DOI:** 10.3390/molecules28104239

**Published:** 2023-05-22

**Authors:** Rodrigo Alonso-Villegas, Rosa María González-Amaro, Claudia Yuritzi Figueroa-Hernández, Ingrid Mayanin Rodríguez-Buenfil

**Affiliations:** 1Facultad de Ciencias Agrotecnológicas, Universidad Autónoma de Chihuahua, Av. Pascual Orozco s/n, Campus 1, Santo Niño, Chihuahua 31350, Chihuahua, Mexico; ralonso@uach.mx; 2CONACYT-Instituto de Ecología, A.C. Carretera Antigua a Coatepec 351, Col. El Haya, Xalapa 91073, Veracruz, Mexico; rosa.gonzalez@inecol.mx; 3CONACYT-Tecnológico Nacional de México/Instituto Tecnológico de Veracruz, Unidad de Investigación y Desarrollo en Alimentos, M. A. de Quevedo 2779, Veracruz 91897, Veracruz, Mexico; 4Centro de Investigación y Asistencia en Tecnología y Diseño del Estado de Jalisco A.C. Subsede Sureste, Tablaje Catastral, 31264, Carretera Sierra Papacal-Chuburna Puerto km 5.5, Parque Científico Tecnológico de Yucatán, Mérida 97302, Yucatán, Mexico

**Keywords:** chili, *Capsicum*, bioactive compounds, polyphenols, biological activity, in vitro evaluation

## Abstract

Chili is one of the world’s most widely used horticultural products. Many dishes around the world are prepared using this fruit. The chili belongs to the genus *Capsicum* and is part of the *Solanaceae* family. This fruit has essential biomolecules such as carbohydrates, dietary fiber, proteins, and lipids. In addition, chili has other compounds that may exert some biological activity (bioactivities). Recently, many studies have demonstrated the biological activity of phenolic compounds, carotenoids, and capsaicinoids in different varieties of chili. Among all these bioactive compounds, polyphenols are one of the most studied. The main bioactivities attributed to polyphenols are antioxidant, antimicrobial, antihyperglycemic, anti-inflammatory, and antihypertensive. This review describes the data from in vivo and in vitro bioactivities attributed to polyphenols and capsaicinoids of the different chili products. Such data help formulate functional foods or food ingredients.

## 1. Introduction

Horticultural products are an essential ingredient of all diets worldwide, as they are a reliable source of carbohydrates, dietary fiber, protein, lipids, and minerals. Additionally, these products contain compounds in a lower concentration that can have some physiological activity or bioactivity in the body when they are regularly consumed [1]. Chili (genus *Capsicum*) is one of the most important horticultural products. In the last two decades, many studies have shown this fruit’s possible biological activities [2,3,4,5,6]. Some of the biological activities that have been evaluated in distinct species of the genus *Capsicum* are the following: (i) antioxidant [7,8,9,10,11,12], (ii) antimicrobial [9,13,14], (iii) anti-inflammatory [15], (iv) antihypertensive [3,16], (v) antihyperglycemic [3,17], (vi) metal-chelating [18,19], and (vii) antitumoral [20].

Chili belongs to the *Solanaceae* family [21,22,23,24,25], which includes eggplant, tomatoes, potato, and tobacco. It is an herbaceous plant or a small shrub with white or pink flowers pollinated by insects such as bees, bumblebees, and aphids. Plants of the genus *Capsicum* can grow worldwide, but their origin is in America. However, the tropical and subtropical environment is optimal for plant growth [22,24,26]. The pre-Columbian distribution of this genus probably extended from the southernmost edge of the United States to the warm temperate zone of southern South America [27]. The genus *Capsicum* contains more than 30 species of chili [28,29]. Still, the five most cultivated and representative species are *Capsicum annuum* L., *Capsicum chinense* Jacq, *Capsicum frutescens* L., *Capsicum baccatum* L., and *Capsicum pubescens* (Ruiz and Pav). The distribution of the genus *Capsicum* is shown in Appendix A. The first four are in Mexico; *C. frutescens* and *C. annuum* have been domesticated there. *C. annuum* is the most cultivated and economically important species in the world. Mexico is its center of diversification, with more than 100 morphotypes present [30,31,32,33]. There is evidence of its cultivation for approximately 6000 years. Its uses date back to pre-Columbian times, where its primary use was as a condiment, but the different types of chili peppers also played an essential role as a source of vitamin C in the various American cultures [34].

According to Market Report World data, the global *Capsicum* market size was valued at USD 4471.04 million by 2022, and the market is expected to reach USD 7325.67 million by 2028, at a CAGR of 8.58% [35]. The world’s leading chili pepper-producing countries in 2021 were China (16,725,716 tons), Turkey (2,864,100 tons), Indonesia (2,759,805 tons), and Mexico (2,701,293 Tons) [36]. Based on data from the Servicio de Información Agroalimentaria y Pesquera (SIAP) reported in 2021, Mexico’s production of chili was 3 million tons, representing 20.2% of the country’s total vegetable production [37]. The most cultivated chilies in Mexico are habanero, jalapeño, serrano, piquin, manzano, and serrano [38,39]. 

The fruits of the genus *Capsicum* contain nutritionally relevant metabolites such as carotenoids (provitamin A), ascorbic acid (vitamin C), tocopherols (vitamin E), phenolic compounds, and capsaicinoids [40,41,42]. These metabolites might have beneficial health effects, and some may exert antioxidant activity because they may have free radical scavenging and oxygen scavenging capabilities [43,44]. A significant percentage of the antioxidant activity that chili has, is due to the number of phenolic compounds and not only to the content of vitamins and carotenoids; therefore, the study of the phenolic fraction is very important [18,45,46]. 

Traditionally, chili is usually consumed fresh and processed in different forms, such as spice, powder, paste, etc. [8,47]. In addition, there are other uses that our ancestors developed, such as in medicine, punishment, currency, and tribute material, among others. It is used in traditional medicine to remedy the effect of asthma, cough, throat irritation, and other respiratory disorders [25,48,49]. This manuscript aims to describe the main bioactivities attributed to the genus *Capsicum,* its polyphenol and capsaicinoid composition, and its potential use in the formulation of functional foods.

## 2. Composition of the Genus *Capsicum*

### 2.1. Nutritional Composition

The genus *Capsicum’s* nutritional composition has been observed to depend on the fruit’s ripeness. *Capsicum* fruit is of ethnopharmacological importance and has been traditionally used in most cuisines and food products due to its distinctive flavor, color, and aroma [2,50,51]. The composition of nutrients and minerals in chili belonging to the genus *Capsicum annuum* L. is shown in Table 1.

Furthermore, to these compounds, chili contains various hydrocarbon chains derived from vanillin amides. These compounds are called capsaicinoids. Capsaicinoids have a linear or branched structure [55]. Other important compounds present in chili are carotenoids (some with provitamin A activity), phenolic compounds, ascorbic acid (vitamin C), and tocopherols (vitamin E) [21,22,56]. The composition and concentration of these metabolites are affected by the ripeness stage, cultivation systems, and fruit processing [22,57,58,59,60,61,62,63].

Some chili metabolites act as defense mechanisms against several abiotic and biotic stresses [22]. It has also been suggested that capsaicin in chili plants forms part of one of the defense mechanisms against frugivorous animals and *Fusarium* [22,57]. These components are valuable not only to the plant but also to humans. On the other hand, phenolic compounds such as cinnamic acids, their derivatives, and flavonoids have in vitro antioxidant activity against free radicals and reactive oxygen species; in addition to potential antitumor activity. In recent years, several studies have been carried out on the bioactive potential of these compounds [22,58]. Moreover, chili has some volatile compounds such as phenols, aldehydes, ketones, ketone alcohols, ethers, nitrogen compounds, aromatic hydrocarbons, alkanes, esters, and lactones [55,59,60]. A brief description of some of the compounds of the *Capsicum* genus (polyphenols and capsaicinoids) that exert bioactivity is given below.

### 2.2. Bioactive Compounds Found in Chili Peppers

Since ancient times, it has been known that chili has a wide range of therapeutic properties. There are several applications for chili extracts in the formulation of various drugs. In the market, several ointments contain capsaicin, which is administered topically for pain relief, migraines, headaches, psoriasis, and herpes simplex virus infection [2,61]. Capsaicin also treats dyspepsia, lack of appetite, flatulence, atherosclerosis, heart disease, and muscle tension [55]. Other bioactive compounds attributed to various chili varieties are shown in Table 2. 

#### 2.2.1. Phenolic Compounds

Phenolic compounds are secondary metabolites that plants synthesize during their growth or in response to stress conditions. These compounds are related to defense mechanisms against radiation and pathogenic microorganisms [58,71,72,73]. Reactive oxygen species (ROS) are known to be responsible for oxidative stress. ROS are involved in developing chronic diseases such as atherosclerosis, cancer, obesity, diabetes, and coronary heart disease [43,74,75]. In the last decade, several studies highlighting phenolic compounds’ protective effects on the progression of these chronic diseases have been carried out. This effect is thought to be due to the antioxidant activity of the phenolic compounds [76,77]. Most studies on the profile of phenolic compounds in species of the genus *Capsicum* are based on the content of capsaicinoids, which are distinctive compounds in pungent chili (*Capsicum annuum* and *Capsicum chinense* Jacq.). However, phenolic compounds have also been studied in low-pungency chili pepper species [78]. The predominant phenolic compounds in chili are catechin, quercetin, protocatechuic acid, and rutin [72,79,80,81,82].

##### Flavonoids

Flavonoids are secondary metabolites of low molecular weight found in fruits, vegetables, herbs, spices, stems, and flowers. They occur in free form or are esterified as glucosides. Flavonoids can exert several crucial pharmacological functions in the body. Although very diverse, the flavonoids characteristically have a benzopyrone skeleton with varying degrees of saturation, and functional groups added to the rings. The most common flavonoids in nature are classified into one of six groups listed below: flavones, flavanones, chalcones, anthocyanins, condensed tannins, and flavonols [83,84,85]. In leguminous plants, other flavonoids known as isoflavones are also synthesized. In nature, there are about 6000 different flavonoids, which have various biological functions such as protection against UV radiation, protection against phytopathogens, signaling during the nodulation process, and coloring of the flowers as a visual signal that attracts pollinators, and protects the leaves by preventing photo-oxidative damage. Flavonols are the most relevant flavonoids and are involved in stress responses. They are also nature’s oldest and most distributed flavonoids [83,86,87].

In general, flavonoids are synthesized through the phenylpropanoid route, which is responsible for transforming phenylalanine to 4-coumaroyl-CoA, which enters the synthesis pathway of flavonoids. This central route for flavonoid biosynthesis is prevalent in plants. However, depending on the species, other enzymes (isomerases, reductases, hydrolases) may act on them to modify their original skeleton and produce differences in the structures of flavonoids [83]. In the genus *Capsicum*, the glycosides of quercetin, luteolin, apigenin, and catechin form part of the composition of flavonoids. Quercetin glycosides are found only in the O-glycosylated form and are the most abundant forms (quercetin 3-O-rhamnoside and quercetin-7-O-rhamnoside). Apigenin glycosides are C- and O-glycosides, while luteolin is only found in its C-glycosylated structure [16]. Luteolin has been reported to have antioxidant, anticancer, anti-inflammatory, and neuroprotective effects [88,89]. The concentration of total flavonoids in chili pepper may vary depending on the cultivar. Some researchers have reported correlations between flavonoid composition and chili color. However, other studies have not reported this relationship [21]. In some varieties of purple or violet chilies, anthocyanins (delphinidin) have been reported as the compounds responsible for the color of such varieties [90].

Quercetin glycosides are found only in the O-glycosylated form and are the most abundant forms (quercetin 3-O-rhamnoside and quercetin-7-O-rhamnoside). Apigenin glycosides are C- and O-glycosides, while luteolin is only found in its C-glycosylated structure [16]. The concentration of total flavonoids in chili pepper may vary depending on the cultivar. Some researchers have reported correlations between flavonoid composition and chili color. However, other studies have not reported this relationship [21]. In some varieties of purple or violet chilies, anthocyanins (delphinidin) have been reported as compounds responsible for the color of such varieties [90].

#### 2.2.2. Capsaicinoids and Capsinoids

Capsaicinoids are amides produced by species of the genus *Capsicum*. These substances are responsible for the spicy and pungent flavor of chili. Capsaicinoids are compounds that differ in structure from the residues of branched fatty acids attached to a benzene ring of vanillylamine. Any variation in the chemical structure of the capsaicinoids, including the acyl moieties, affects the degree and level of pungency [22,91,92,93,94]. Capsaicinoids include capsaicin, dihydrocapsaicin, nordihydrocapsaicin, and homocapsaicin. The main capsaicinoids found in hot chilies are capsaicin and dihydrocapsaicin (they may represent 90% of total capsaicinoids), whereas norhydrocapsaicin, homodihydrocapsaicin, and homocapsaicin are found in low concentrations. The capsaicinoids are generally expressed in Scoville Heat Units (SHU). This scale is used to measure the degree of the pungency of chili. The number of Scoville Heat Units indicates how many times a chili extract must be diluted to make its pungency imperceptible. The scale of Scoville categorizes in four groups the pungency degree of chilies is non-pungent (0–700 SHU), mildly pungent (700–3000 SHU), pungent (3000–25,0000 SHU), highly pungent (25,0000–70,000 SHU), and extremely pungent (>80,000 SHU) [73,95]. The chili with the highest pungency is the Trinidad Moruga Scorpion, with up to 2,000,000 SHU [96,97,98].

Capsaicin is synthesized in the chili placenta by enzymatic condensation of vanillylamine with a medium chain branched fatty acid. The key enzyme involved in the formation of capsaicinoids is capsaicin synthetase (CS). This enzyme is an acyltransferase responsible for the condensation of vanillylamine with a fatty acid. The levels of capsaicinoid production and their relative abundance vary depending on the cultivar [99,100,101].

Capsinoids, analogous to capsaicin, have been found in bell pepper (*Capsicum*). The main capsinoids of bell pepper are capsiate, dihydrocapsiate, and nordihydrocapsiate [102,103,104]. These substances share a structure similar to capsaicinoids as they have an aliphatic hydroxyl group in a vanillin alcohol bound to a fatty acid [99,105]. The difference lies in the type of bonds because capsaicinoids have amide bonds, while capsinoids have ester bonds [103,106,107,108,109]. 

Over the last fifteen years, capsaicinoids and capsinoids have gained much interest because they can exert physiological effects when administered at specific concentrations (bioactivity). These bioactivities are increasingly relevant in the pharmaceutical sector, as they are used in the formulation of ointments and medicines. The main bioactivities that have been attributed to capsaicinoids and capsinoids are the following: (i) analgesic activity, (ii) anticancer activity, (iii) anti-inflammatory activity, and (iv) anti-obesity activity [99,110,111,112,113,114].

## 3. Bioactivities Associated with Polyphenols and Capsaicinoids of the Genus *Capsicum*

As mentioned above, chili not only has nutritional compounds but also contains bioactive compounds. To date, the most studied bioactivities in several types of chilies and peppers are antioxidant and antimicrobial activity [8,9,11,14,62,115]. More recently, studies with other bioactivities, such as antihyperglycemic and antihypertensive activity, have been given greater importance because natural alternatives are sought to help control two of the world’s most severe public health problems: hypertension and diabetes [3,116,117,118]. The bioactivities studied in chili extracts, mainly attributed to polyphenols and capsaicinoids, will be briefly described below.

### 3.1. Antioxidant Activity

The most studied biological activity in fruits of the genus *Capsicum* is antioxidant activity. This activity can be quantified in vitro by several methodologies, among which the following are highlighted: antioxidant activity by uptake of the 1,1-diphenyl-2-picrylhydrazyl radical (DPPH), antioxidant activity by uptake of the 2,2′-azino-bis-(3-ethylbenzothiazoline-6-sulphonic acid) radical cation (ABTS); oxygen radical absorption capacity (ORAC); ferric reducing-antioxidant power (FRAP); cupric reducing antioxidant capacity (CUPRAC); β-carotene bleaching assay; superoxide radical scavenging activity (SOD); thiobarbituric acid method (TBA); hydroxyl radical averting capacity (HORAC); reducing power method (RP), and the ferric thiocyanate method (FTC) [119,120,121,122,123,124]. In most of these studies, the concentration of polyphenols, capsaicinoids, and other compounds, such as carotenoids, is directly related to the antioxidant activity present in chili and peppers. 

Hervert-Hernández et al. [8] evaluated the antioxidant activity of the polyphenolic extracts of four varieties of hot chilies. The hot peppers analyzed were Arbol, Chipotle, Guajillo, and Morita. The antioxidant activity was quantified using two ABTS and FRAP methods. The authors found that the total antioxidant activity of the chilies quantified by the FRAP method was 63.9 and 82.3 μmol of Trolox equivalents/g of dry matter; the Guajillo chili was the only one that presented the lowest antioxidant activity. In the case of total antioxidant activity evaluated using the ABTS method, the highest activity was found in the Chipotle chili (44 ± 0.6 μmol Trolox equivalents/g of dry matter), and the lowest activity was observed in the Guajillo chili (26.6 ± 1.0 μmol Trolox).

Galvez-Ranilla et al. [65] quantified the antioxidant potential of extracts of medicinal plants, chilies, and some herbs widely used in Latin America. The analyzed extracts corresponding to the genus *Capsicum* were: Arbol, Ancho, Yellow, Japanese, Red, Paprika, and Rocoto. It was found that all extracts analyzed showed antioxidant activity, which was measured using the DPPH method (between 61 and 73%). Additionally, it was found that red chili had the highest antioxidant activity (73%). The antioxidant activity found in the extracts of the chilies evaluated in this study was due to the concentration of polyphenols and other compounds, such as carotenoids. Another work carried out by Ghasemnezhad et al. [66] determined the concentration of phenolic compounds, ascorbic acid, and the antioxidant capacity of extracts of various red chilies (*Capsicum annuum*) in two different maturity stages. The authors found that the higher antioxidant potential (measured by the DPPH method) depended on the type of variety and the maturity stage because chili peppers with higher maturity showed higher antioxidant activity.

Zhaung et al. [67] analyzed the bioactive compounds (vitamin C, carotenoids, phenolic compounds, and capsaicinoids) and the antioxidant activity of nine pepper cultivars from Yunnan Province in China. The antioxidant activity of pepper cultivars were evaluated using DPPH, reducing power, and lipid peroxidation inhibitory activity. The ethanolic extracts of chili cultivars showed significant antioxidant activity (quantified using three methods). However, the red Fructus Capsici variety extracts had a higher antioxidant potential and a higher concentration of phenolic compounds. It was also observed that the antioxidant activity of all extracts evaluated was directly related to phenolic compounds content.

In a study conducted by Segura-Campos et al. [11], the antioxidant activity of the Habanero chili was analyzed using the β-carotene decolorization method and the ABTS method. The authors used seven genotypes of Habanero pepper (*Capsicum chinense* Jacq.) from the state of Yucatan for the study. The antioxidant activity reported as Trolox equivalents (TEAC) ranged from 1.55 to 3.23 mM/mg sample. For antioxidant activity quantified as β-carotene bleaching, it was observed that the extracts could bleach the β-carotene between 36 and 57% during the first 30 min. The authors suggest that the antioxidant activity is related to the phenolic compounds, carotenoids, and ascorbic acid concentration.

Another work by Castro-Concha et al. [10] quantified the antioxidant potential and content of total polyphenols of red and orange Habanero peppers (*Capsicum chinense* Jacq.) at different stages of maturity. The extracts of the peppers were obtained from two different stages of maturity (immature and mature). The antioxidant activity was evaluated using the DPPH method and the cupric-reducing antioxidant capacity (CUPRAC). The immature placental tissue of the red habanero pepper was the one that presented the highest antioxidant activity quantified in both methods, and it also showed the highest number of phenolic compounds.

Carvalho et al. [7] quantified the concentration of bioactive compounds (polyphenols, anthocyanins, carotenoids, and vitamin C) and the antioxidant activity of various chili genotypes. The genotypes used in this study belong to the species *Capsicum sp*., *Capsicum annuum* L., *Capsicum chinense* Jacq., and *Capsicum baccatum* L. var. *umblicatum*. The antioxidant activity of the chili extracts (methanol and acetone) was measured using the ABTS and DPPH methods. It was found that genotype IAN186311 showed the highest antioxidant activity and the highest concentration of phenolic compounds.

In another work, Chavez-Mendoza et al. [68] studied the antioxidant potential by selecting different combinations of varieties and rootstock of various bell pepper cultivars (*Capsicum annuum* L.). The combinations used in this study were: Jeanette/Terrano (yellow), Sweet/Robusto (green), Fascinato/Robusto (red), Orangela/Terrano (orange), and Fascinato/Terrano (red). The antioxidant activity of the extracts was determined by the DPPH method. The extract of the combination Fascinato/Robusto was the extract that presented higher antioxidant activity (79.65%). The extract with the Jeanette/Terrano combination showed the lowest antioxidant potential (64.90%) of DPPH radical inhibition. This study demonstrated that antioxidant activity was significantly affected by the cultivar/rootstock combination and the color of the peppers used. The authors observed that the Fascinato/Robusto combination had the highest concentrations of lycopene and total phenols and the most increased antioxidant activity of all cultivar/rootstock combinations evaluated.

Loizzo et al. [69] evaluated the antioxidant potential of twenty chili cultivars belonging to *Capsicum annuum*, *Capsicum baccatum*, *Capsicum chacoense*, and *Capsicum chinense*. This study assessed the antioxidant activity of fresh and processed chili extracts. The antioxidant activity of the ethanolic extracts of chilies was evaluated using four methods: ABTS, DPPH, FRAP, and β-carotene bleaching. The cultivars of the species *Capsicum annuum* showed the highest antioxidant activity. Nevertheless, the researchers reported that the antioxidant activity of chili peppers does not directly correlate with the concentration of phytochemicals they contain, demonstrating that different classes of compounds are poorly associated with antioxidant parameters, even within assays in which they share the same mechanism of action.

Sora et al. [125] conducted a comparative study of the antioxidant activity, capsaicinoid content, and polyphenols found in six extracts of chili pepper extracts belonging to the genus *Capsicum*. The extracts analyzed were from Bell, Habanero orange, Cayenne, Habanero red, Malagueta, and Dedo de moça chili peppers. To quantify the antioxidant activity of the ethanolic extracts of seeds and pulps of chili used. Antioxidant activity was measured using ABTS, FRAP, and DPPH assays. The antioxidant activity was higher in the seeds than in the pulp. This bioactivity was directly related to the content of polyphenols and capsaicinoids. The ethanolic extract of red habanero seeds showed the highest antioxidant activity. 

The antioxidant activity of the capsaicinoids and carotenoids extracted from Algerian chili was evaluated using the DPPH, ABTS, and FRAP methods. The carotenoid extracts showed higher antioxidant activity than capsaicinoids. The antioxidant activity of the carotenoid extract was: 2.8 ± 0.3 mmol Trolox equivalents/g determined using the DPPH method, 3.4 ± 0.003 mmol Trolox equivalents/g determined using the ABTS method, and 5.9 mmol Fe^2+^/g determined using the FRAP method [9].

Hamed et al. [73] studied the capsaicinoids and other bioactive compounds in chilies from different cultivars at two maturity stages (green and red) on antioxidant activity. The effect of roasting on their nutritional content was also investigated. They found that both raw and roasted chilies exhibited strong antioxidant activity as determined by DPPH (61–87%) and ABTS (73–159 μg/g). The results showed a relatively weak positive correlation of total phenolics and total flavonoid concentration with DPPH and ABTS antioxidant activities, with the highest correlation between total phenolics concentration (r = 0.55) and the DPPH antioxidant assay.

In 2020, Oney-Montalvo et al. [80] correlated the total polyphenol content and the main phenolic compounds (catechin, chlorogenic acid, ellagic acid, gallic acid, and protocatechin) of habanero peppers with their antioxidant activity (measured by ABTS and DPPH assays). The results indicated that the total polyphenol content and antioxidant activity of chiles showed a correlation of r^2^ = 0.8999 (DPPH) and r^2^ = 0.8922 (ABTS). In addition, they found a good correlation of antioxidant activity with catechin (r^2^ = 0.8661 (DPPH) and r^2^ = 0.8989 (ABTS)), chlorogenic acid (r^2^ = 0.8794 (DPPH) and r^2^ = 0. 8934 (ABTS)) and ellagic acid (r^2^ = 0.8979 (DPPH) and r^2^ = 0.9474 (ABTS)), indicating that these polyphenols contribute significantly to the antioxidant activity found in habanero peppers. 

### 3.2. Antimicrobial Activity

The microbiological quality of food is one of the major concerns of the food sector. There are currently many strategies for controlling microbial growth in food; however, some issues still need to be resolved. Today, many chemical preservatives are approved for use in food, but the current trend is the use of naturally occurring antimicrobial substances, which are safe, effective, and sensorial accepted [126,127]. A study by Mokhtar et al. [9] evaluated the antimicrobial potential of an extract of capsaicinoids from Algerian chili (*Capsicum annuum*). It was found that this extract had activity against *Listeria monocytogenes* ATCC 1392 and *Enterococcus hirae* ATCC 10541. Additionally, this same study proved that this extract had no antimicrobial activity against the beneficial bacteria *Lactobacillus rhamnosus* LbRE-LSAS and *Bifidobacterium longum* ATCC15707. 

Nascimento et al. [13] quantified and evaluated the antimicrobial potential of capsaicin, dihydrocapsaicin, and chrysoeriol extracted from different tissues (fruit, seeds, and shell) of various Malagueta chili varieties. The antimicrobial effect was evaluated against the growth of the pathogenic microorganisms: *Enterococcus faecalis*, *Bacillus subtilis*, *Staphylococcus aureus*, *Pseudomonas aeruginosa*, *Klebsiella pneumoniae*, *Escherichia coli*, and *Candida albicans*. The minimum inhibitory concentration of each extract was determined against each pathogenic microorganism, and the three compounds showed a significant antimicrobial effect against the microorganisms evaluated in this study (0.06 to 25 μg/mL). Capsaicin, dihydrocapsaicin, and chrysoeriol inhibit the growth of Gram-positive and harmful bacteria.

Gayathri et al. [62] determined the antimicrobial potential of capsaicin extracts (acetone and acetonitrile) obtained from various tissues (callus, leaves, shoots, fruits, and seeds) of *Capsicum chinense* Jacq. The obtained extracts have antimicrobial activity against *Salmonella typhi*, *Aspergillus flavus*, *Bacillus cereus*, *Staphylococcus aureus*, and *Streptococcus pyogenes*.

### 3.3. Anti-Inflammatory Activity

The inflammation process is a natural defense mechanism of the body’s immune system in response to damage caused to cells and tissues by toxic agents such as microorganisms, chemicals, and necrosis. Mostly, it is a protective response that localizes and destroys the injurious agent and then prepares the damaged tissue for repair. However, inflammation and oxidative stress are involved in the development of various diseases such as cancer, rheumatoid arthritis, asthma, diabetes, and cardiovascular and degenerative illness [15,63].

In a study by Spiller et al. [63], the anti-inflammatory potential of red chili (*Capsicum baccatum*) has been quantified on an inflammation induced by carrageenan and on an immune inflammation induced with bovine serum albumin methylated in mice. It was observed that pretreatment with red chili juice at a concentration between 0.25 and 2 g/kg applied 30 min before carrageenan administration significantly reduces leukocyte and neutrophil migration, exudate volume, protein concentration, and the level of lactate dehydrogenase (LDH) in the exudates of the induced pleurisy model. Red chili juice also inhibits the migration of neutrophils and reduces vascular permeability in carrageenan-induced peritonitis in rats. On the other hand, this extract reduces the recruitment of neutrophils and levels of pro-inflammatory cytokines (TNF-α and IL-1β) in immune-induced peritonitis in rats. The authors suggest these effects are due to the capsaicin in red chili juices.

Zimmer et al. [15] evaluated the antioxidant and anti-inflammatory potential of red chili (*Capsicum baccatum*) seeds and pulp extracts. The extracts were extracted with ethanol, butanol, and dichloromethane. Anti-inflammatory activity was quantified using carrageenan-induced pleurisy models in mice. This study observed an anti-inflammatory effect of ethanolic and butanolic extracts (200 mg/kg per os) compared to dexamethasone (0.5 mg/kg subcutaneously). It was also observed that the content of polyphenols might be related to the antioxidant and anti-inflammatory activity of the extracts evaluated.

Cho et al. [64] studied the in vitro anti-inflammatory effects and flavonoid content of pepper leaves (PL) and pepper fruit (FP). Pepper extracts ameliorated the lipopolysaccharide (LPS)-stimulated inflammatory response by decreasing nitric oxide production and the levels of interleukin-6 and tumor necrosis factor-alpha in RAW 264.7 cells, with greater efficacy of the activities of PL than FP.

Simpson et al. [128] studied the effect of a single application of a high-concentration capsaicin dermal patch (NGX-4010) in patients with HIV-associated distal sensory polyneuropathy. The patch application was safe and provided at least 12 weeks of pain reduction in patients. Thus, the authors suggest that these results could be used as a promising new treatment for painful HIV neuropathy.

### 3.4. Antihypertensive Activity

Angiotensin-converting enzyme (ACE) is a crucial enzyme in the control of blood pressure. This enzyme converts angiotensin I (inactive decapeptide) to angiotensin II (vasoconstrictor octapeptide) and hydrolyzes bradykinin, a potent vasodilator, forming inactive fragments. Inhibition of ACE activity is a therapeutic alternative for the control of hypertension [129]. Currently, food-derived ACE-inhibitory compounds are being sought to treat hypertension, as it is one of the most serious health problems worldwide. ACE-inhibition is one of the methods used to quantify antihypertensive activity in vitro. 

Galvez-Ranilla et al. [65] studied the ACE-inhibitory activity of the extracts of medicinal plants, herbs, chilies, and species commonly used in Latin America. The inhibitory potential of the angiotensin-converting enzyme (ACE) from chili extracts was evaluated at three different concentrations (0.5, 1.25, and 2.5 mg dry weight). It was found that almost all extracts of the evaluated chilies showed ACE- inhibitory activity except the extracts at a concentration of 0.5 mg of Arbol, Ancho, and Rocoto chili peppers. The ACE-inhibitory activity of the analyzed extracts was between 20 and 90%. They found a correlation (r = 0.61) between ACE-inhibitory activity and polyphenol concentration.

In another work, Chen and Kang [3] evaluated the inhibitory potential of essential enzymes in controlling hypertension and hyperglycemia from extracts of various tissues of red chili (*Capsicum annuum* L.). The evaluated tissues were the pericarp, placenta, and stalk. The ACE-inhibitory potential of methanolic tissue extracts was tested at three different concentrations (1, 3, and 5 mg/mL). The extracts obtained from the pericarp showed the highest ACE-inhibitory activity. The degree of inhibition depended on the concentration of the extract. Regarding the ACE-inhibitory potential of the placenta extracts, they only inhibited the enzyme when the extract had a concentration of 3 and 5 mg/mL. The extracts obtained from the stalk of red chili only inhibited the enzyme’s activity by 10% when they had a concentration of 5 mg/mL. On the other hand, in this work, the high total phenolic content did not correlate to the ACE-inhibitory activity of these extracts. However, this activity is unrelated to the concentration of polyphenols found in the evaluated extracts.

### 3.5. Antihyperglycemic Activity

Diabetes is one of the significant health problems worldwide. Type 2 diabetes is a metabolic disorder characterized by a high glucose concentration in the blood due to deficiency or null insulin production. In patients with diabetes, levels of α-amylase and α-glucosidase may be harmful. Inhibition of these enzymes may reduce the increase in blood glucose after food intake; and therefore, these can be a good alternative for the control of the level of blood glucose. The search for naturally occurring compounds that can control blood glucose levels for patients with type 2 diabetes is in process. Inhibition of α-amylase in the small intestine has been observed to be related to blood glucose levels after food intake; hence, the control of the activity of this enzyme is a crucial factor for the treatment of diabetes. Plant extracts are currently explored as natural alternatives to control hyperglycemia as an option to existing pharmacological treatments. The antihyperglycemic effect can be assessed in vitro by inhibition of α-amylase and α-glucosidase [3].

In a study conducted by Galvez-Ranilla et al. [65], the antihyperglycemic activities of extracts of spices, chilies, medicinal plants, and herbs were evaluated. The extracts from the chilies analyzed were: Arbol, Ancho, Yellow, Japanese, Red, Paprika, and Rocoto chili. The extracts were evaluated for three concentrations 0.5, 1.25, and 2.5 mg. The antihyperglycemic activity of the extracts was assessed by the inhibition of α-amylase and α-glucosidase. All the extracts analyzed had α-glucosidase inhibitory activity. However, only the extracts analyzed at a concentration of 1.25 mg had α-amylase inhibitory activity. The highest inhibitory activity of α-glucosidase was close to 50% in the Red, Rocoto, and Yellow chili extracts. Yellow, red, paprika, and Japanese chili pepper showed an α-amylase inhibitory activity close to 40%. However, this activity depends not only on the concentration of total phenols in chili but is due to a synergistic effect between these and other compounds in chili, such as capsaicinoids and carotenoids.

Tundis et al. [70] evaluated the antihyperglycemic activity and its relationship with phytochemicals in cultivars of *Capsicum annuum* during its development. The varieties of chili assessed in this study were: Fiesta, Acuminatum, Orange Thai, and Golden Cayenne. The antihyperglycemic activity was evaluated by the enzymatic inhibition of α-amylase and α-glucosidase. The ethanolic extracts of the immature Fiesta pepper, Orange Thai and Golden Cayenne were the ones that showed a more significant inhibitory effect of α-amylase. Regarding the α-glucosidase inhibitory activity, all ethanolic extracts of the mature and immature chilies evaluated were revealed to have a mean inhibitory activity (IC_50_ ≤ 166.5 μg/mL), except the extracts of Fiesta and Acuminatum chili peppers in the mature state (IC_50_ > 1000 μg/mL). The concentration of phenols and flavonoids was higher in the immature chilies, while the concentration of carotenoids and capsaicinoids was higher in the mature chilies. In contrast, the lipophilic fraction of the chili extracts at the two maturity stages evaluated showed a high inhibitory effect on the activity of the α-amylase enzyme (IC_50_ ≤ 30 μg/mL) [69]. In addition, the authors found that both capsaicin and dihydrocapsaicin showed an inhibitory effect on α-amylase with IC_50_ of 83.0 and 92.0 μg/mL, respectively.

Chen and Kang [3] quantified the inhibitory potential of essential enzymes involved in hyperglycemia from extracts of various tissues of red chili. The inhibitory activities of α-amylase and α-glucosidase were evaluated using three different concentrations of extracts (1, 3, and 5 mg/mL). The methanolic extracts of the pericarp showed the highest degree of inhibition of α-amylase. It was also observed that the α-amylase inhibitory activity was unrelated to the polyphenol concentration. On the other hand, all the extracts evaluated at the three different concentrations showed α-glucosidase inhibitory activity. Methanolic red chili pericarp extracts showed higher inhibitory activity (≥30%). This study observed that the degree of inhibition of both enzymes depends on the dose of the extract used. The authors observed that high polyphenol contents correlated with higher alpha-glucosidase inhibitory activity for cultivar B.

### 3.6. Metal-Chelating Activity

Metal-chelating activity is intrinsically related to antioxidant activity. It is known that the binding of minerals such as iron and copper has an antioxidant effect because these metals can cause oxidative damage in cells at different levels. It has been observed that these oxidative reactions are involved in the pathogenesis of some neurodegenerative diseases [130].

Siddiqui et al. [19] studied the dynamics of changes in bioactive molecules and the antioxidant potential of the extracts of Habanero pepper during nine different states of maturity. The antioxidant activity was determined using the DPPH method and the metal-chelating activity (chelating of ferrous ions). The authors concluded that the iron chelating activity of ethanolic extracts increases as a function of maturity until reaching a maximum. The results indicated that at day 49 postharvest, extracts reached 75% iron-chelating activity. Furthermore, the authors observed that metal-chelating activity was strongly correlated with the concentration of total polyphenols (r = 0.972) and total flavonoids (r = 0.956).

### 3.7. Antitumoral Activity

A tumor can be defined as the uncontrolled multiplication of cells; the tumors can be benign or malignant. If a tumor is mildly benign, the cells only multiply uncontrollably but do not spread to another part of the body. Usually, this type of tumor does not endanger life. A malignant tumor or cancer can spread to other organs (metastasis). Cancer is one of the most significant health challenges worldwide. It is currently the second leading cause of death in the United States of America and is expected to become the leading cause of death in that country in the next few years [131,132].

Mokhtar et al. [9] evaluated the antitumoral potential of capsaicinoids and carotenoids extracted from Algerian chili (*Capsicum annuum*) against cancerous (U937) and healthy (PBMC) cell lines. Cell lines were exposed to various concentrations of extracts to observe their viability. The effect of the two extracts was higher in cancer cells than in healthy PBMC cells. The extracted capsaicinoids had a lower antitumoral effect than carotenoids. However, the antitumor activity of capsaicinoids at a concentration of 200 μg/mL was 52%, whereas, for carotenoids, antitumoral activity reached 90% activity at the same concentration.

Wang et al. [133] studied the effect of capsaicin on gastric cancer (GC) cell lines to understand the mechanism of cell growth inhibition. The researchers demonstrated that capsaicin could significantly suppress cell growth while altering histone acetylation in GC cell lines. In this study, reduced hMOF activity (histone H4 lysine K16-specific acetyltransferase) was detected in GC tissues, which capsaicin could restore in both in vivo and in vitro studies. These findings revealed an important role of hMOF-mediated histone acetylation in capsaicin-directed anticancer processes, and therefore, the authors suggested capsaicin as a potential drug for gastric cancer prevention and therapy.

## 4. Incorporation of Polyphenols and Capsacinoids from *Capsicum* on Food and Cosmetics Products 

Chili paste is a fermented product typical of China. During fermentation, the burning sensation caused by eating unprocessed chili peppers is mitigated. In addition, it is a technique to preserve food and has unique sensory attributes [134]. Several studies have been conducted with chili paste fermented with different strains of lactic acid bacteria and yeasts [134,135,136,137,138,139]. However, most of these studies have studied the effect on the sensory attributes produced by fermentation [135,136,137,138] or monitoring of the microorganisms during autochthonous chili fermentation [139]. Consequently, further studies are needed to study the effect of fermentation on bioactive compounds (mainly phenolic compounds). In this regard, it is important to note that certain strains of LAB and yeast, during fermentation, can metabolize phenolic acids and thus increase their bioactivity or bioavailability [140,141,142,143,144,145,146,147]. This can lead to an increase in antioxidant, anti-inflammatory and antiglycemic activity, among others [146,147,148]. Therefore, studies on the effect of fermentation with LAB or yeast strains on the concentration of polyphenols, as well as on the bioactive effects due to the microbial transformation of these compounds, will be of great scientific importance. 

Furthermore, in 2014, Song et al. [147] developed a vinegar with chili leaves (*Capsicum annuum* L.) that exhibited enhanced functional activity. The production of the vinegar supplemented with pepper leaves was carried out following a four-stage procedure: (i) preparation of the raw materials, (ii) a lactic fermentation (*Fructilactobacillus fructivorans* formerly known as *Lactobacillus homohiochii*), (iii) an alcoholic fermentation (*Saccharomyces cerevisiae*) and (iv) an acetic fermentation (*Acetobacter aceti*). The utilization of fermentation of chili leaves was very effective in producing a functional food by enhancing the content of bioactive components such as gamma-aminobutyric acid (GABA), citrulline, and polyphenols. In addition, the vinegar with chili leaves showed efficacy in the elimination of free radicals and the inhibition of α-glucosidase activity. These bioactivities were attributed to the increase in phenolic concentration during fermentation. Therefore, the authors conclude that the consumption of this vinegar supplemented with chili leaves could be used as a dietary strategy to prevent oxidative stress and alleviate postprandial hyperglycemia in diabetes. 

Zellama et al. [149] evaluated the effect of supplementing extra virgin olive oil (EVOO) with chili. In this study, a higher concentration of polyphenols and ortho-diphenols was found. Four colorimetric methods were used to determine antioxidant activity (DPPH, ABTS, FRAP, and the β-carotene bleaching test). Compared to the non-supplemented control oil, a higher level of antioxidant activity was observed in the chili-enriched oil. In addition, a significant antimicrobial effect was obtained mainly against Gram-positive bacteria. These results provide a scientific basis for innovation in food technology for new food products. In addition, the incorporation of several *Capsicum*-derived ingredients has been documented in a wide variety of commercial products, as shown in Table 3. This shows the importance of the incorporation of these ingredients derived from the *Capsicum* genus in the cosmetic and food industries. On the other hand, it is worth mentioning that in traditional Asian and Mexican foods, many chili peppers are used as part of the recipes of a large number of dishes. However, there are no studies on the effect of chili as an ingredient.

## 5. Conclusions

The genus *Capsicum* is one of the most important horticultural products in the world. Horticultural products are an essential ingredient in all diets. They are a reliable source of carbohydrates, dietary fiber, proteins, lipids, and minerals. This fruit is traditionally used in many dishes and food products for its distinctive flavor, color, and aroma compounds. Additionally, it contains some compounds that may have beneficial effects when consumed on a regular intake, such as polyphenols, capsaicinoids, carotenoids, vitamin C, and tocopherols. Antioxidant, antimicrobial, anti-inflammatory, antihypertensive, antihyperglycemic, metal chelating, and antitumoral activity have been associated with polyphenol and capsaicinoid content. These bioactivities highlight the importance of chili peppers as part of the human diet and as a potential ingredient for the formulation and development of functional foods. Additionally, fermentation of chili pastes with various strains of LAB and yeasts can increase antioxidant, anti-inflammatory, and antiglycemic activities, among others, by the biotransformation of phenolic compounds with the microbial strains. Therefore, this increase in bioactivities can be very important for the formulation of new functional products.

## Figures and Tables

**Table 1 molecules-28-04239-t001:** The average composition of chili belonging to the genus *Capsicum annuum* L. Source: Adapted from [52,53,54].

Nutrient	Quantity ^1^
Water	91.0–92.2 g
Carbohydrates	5.10–6.03 g
Proteins	0.99–1.30 g
Fats	0.30 g
Fiber	1.40–2.10 g
Vitamin A	157–300 mg
Vitamin B1	0.03–0.05 mg
Vitamin B2	0.05–0.08 mg
Vitamin B3	0.98 mg
Vitamin B5	0.20–0.32 mg
Vitamin B6	0.29 mg
Vitamin B12	0.45 mg
Vitamin C	120–128 mg
Sulfur	17 mg
Calcium	7–9 mg
Chlorine	37 mg
Copper	0.017–0.100 mg
Phosphorus	23–26 mg
Iron	0.43–0.50 mg
Magnesium	11–12 mg
Manganese	0.11–0.26 mg
Potassium	211–234 mg
Sodium	4–58 mg
Iodine	0.001 mg

^1^ per 100 g of edible portion.

**Table 2 molecules-28-04239-t002:** Main bioactivities associated with different varieties of chili peppers of the genus *Capsicum*.

Bioactivity	Bioactive Compounds	Capsicum Varieties	Concentration Studied	Models/Cell Lines	Reference
Antimicrobial activity	Capsaicinoids and carotenoids	Algerian chili pepper (*Capsicum annuum* L.)	Capsaicinoids (pericarp) 68.3 µg·g^−1^; (placenta) 754.4 µg·g^−1^; carotenoids (fruit) 1620 µg·100 g^−1^	*Staphylococcus aureus*; *Listeria* *Monocytogenes*; *Enterococcus hirae*	[9]
	Phenols, capsaicinoids, and chrysoeriol	Various Malagueta chili peppers *(Capsicum frutescens)*	Capsaicinoids 109.8 mg·g^−1^; dihydrocapsaicinoids 42.0 mg·g^−1^; chrysoeriol 5.50 mg·g^−1^	Gram-positive bacteria (25 µg·mL^−1^); Gram-negative bacteria (10 µg·mL^−1^); Yeast (25 µg·mL^−1^)	[13]
	Chlorophyll and carotenoids	Various tissues (callus, leaves, shoots, fruits, and seeds) of *Capsicum chinense* Jacq.	Chlorophyll 0.105 mg·g^−1^ and Carotenoids 4.10 mg·g^−1^	Minimal inhibitory concentration (MIC): 5-21 mm inhibitory effect	[62]
Anti-inflammatory activity	Capsaicin	Red chili pepper (*Capsicum baccatum*)	Red pepper juice 0.25–2.0 g·kg^−1^	Carrageenan-induced pleurisy in mice model; Carrageenan-induced peritonitis in mice model	[63]
	Capsaicin and quercetin Flavones and flavonols	Red chili pepper (*Capsicum baccatum*)	Butanol extract from fruit pepper (200 mg·kg^−1^ p.o.)	Carrageenan-induced pleurisy model in mice	[15]
		Pepper extracts (*Capsicum annuum*)	Pepper extracts on IL-6 and TNF-α production in LPS-induced RAW 264.7 cells	Pepper leaves and pepper fruit in vitro assays	[64]
	Phenolic compounds (flavonoids) and capsaicin	Red pepper (*Capsicum annuum* L.)	Total extract (IC50): 287 µg·mL^−1^ mature; lipophilic fraction (IC50): 655 µg·mL^−1^ (mature)	Mature and immature fruit peppers	[17]
	Phenolic compounds and carotenoids	Hot peppers of Arbol, Chipotle, Guajillo, and Morita (*Capsicum annuum* L.)	Arbol pepper 82.3 µmol·g^−1^ dry matter (phenolics) and 106.6 mg·100 g^−1^ dry pepper (carotenoids); Chipotle pepper 44.4 µmol·g^−1^ dry matter (phenolics) chipotle pepper	In vitro enzyme digestion (bioaccessibility)	[8]
	Phenolic compounds (flavonoids)	Peppers: Arbol, Ancho, Yellow, Japanese, Red, Paprika, and Rocoto (*Capsicum annuum, baccatum, chinense, and pubescens*).	Chile de arbol (14.0 mg·g^−1^ dry weight); chile ancho and Japanese chili (14.5 mg·g^−1^ dry weight); paprika pepper (15.0 mg·g^−1^ dry weight); yellow pepper (13.0 mg·g^−1^ dry weight); red pepper (20.0 mg·g^−1^ dry weight); rocoto (12.5 mg·g^−1^ dry weight)	In vitro enzyme analysis	[65]
	Phenolic compounds (flavonoids)	Various red chili peppers (*Capsicum annuum* L.)	Arian (mature 8.60% and ripe 21.50% inhibition); Marona (mature 14.80% and ripe 19.60% inhibition); Zorro (mature 9.80% and ripe 14.80% inhibition)	Harvest times based on maturity stage on phenolic compounds of five different colored Capsicum genotypes	[66]
Antioxidant activity	Phenolic compounds, capsaicinoids, and carotenoids	Nine chili cultivars from Yunnan Province in China (*Capsicum frutescens* L. *and annuum* L.)	Fructus Capsici (IC_50_ = 135.13 µg·mL^−1^) Point pepper (IC_50_ = 233.33 µg·mL^−1^) Long-Point pepper (red) (IC_50_ = 190.70 µg·mL^−1^) Point-pepper (IC_50_ = 286.76 µg·mL^−1^) Long-point-pepper (green) (IC_50_ = 223.33 µg·mL^−1^) Sweet pepper (IC_50_ = 366.67 µg·mL^−1^) Longline pepper (IC_50_ = 283.33 µg·mL^−1^) Screw pepper (IC_50_ = 195.00 µg·mL^−1^) Creasing pepper (IC_50_ = 210.10 µg·mL^−1^)	Antioxidant compositions of nine peppers from Yunnan in China	[67]
	Phenolic compounds and carotenoids	Habanero chili pepper (*Capsicum chinense* Jacq. *var.*)	L-36 (TEAC = 3.23 mM·mg^−1^ sample) L-110 (TEAC = 2.74 mM·mg^−1^ sample) Orange (TEAC = 2.42 mM·mg^−1^ sample) L-184 (TEAC = 1.94 mM·mg^−1^ sample) Red (TEAC = 3.05 mM·mg^−1^ sample) L-149 (TEAC = 1.99 mM·mg^−1^ sample) L-37 (TEAC = 1.55 mM·mg^−1^ sample)	The fruit of seven *Capsicum chinense* Jacq. var. Habanero genotypes grown in Yucatan, Mexico	[11]
	Phenolic compounds	Red and Orange Habanero chili peppers (*Capsicum chinense*)	Chak k’an-iik immature pericarp (4.17 TEAC µmols TE·g^−1^) Chak k’an-iik immature placent (30.08 TEAC µmols TE·g^−1^) Chak k’an-iik mature pericarp (8.84 TEAC µmols TE·g^−1^) Chak k’an-iik mature placent (41.64 TEAC µmols TE·g^−1^) MR8H immature pericarp (4.22 TEAC µmols TE·g^−1^) MR8H immature placent (55.59 TEAC µmols TE·g^−1^) MR8H mature pericarp (6.67 TEAC µmols TE·g^−1^) MR8H mature placent (42.28 TEAC µmols TE·g^−1^)	Fruits tissues of two *Capsicum chinense* accessions	[10]
	Phenolics compounds (anthocyanins), carotenoids, and vitamin C	Various chili genotypes. *Capsicum* sp., *Capsicum annuum* L., *Capsicum chinense* Jacq., and *Capsicum baccatum* L. *var*. *Umblicatum*	Biquinho (IAN 186313) 49.56 µM trolox·g^−1^ Curuçazinho (IAN 1836309) 58.36 µM trolox·g^−1^ olho de mutum (IAN 186324) 77.99 µM trolox·g^−1^ Amarcia (IAN 186312) 62.39 µM trolox·g^−1^ Cumari do Pará (IAN 186310) 46.79 µM trolox·g^−1^ PMO (IAN 186301) 83.59 µM trolox·g^−1^ Murupi (IAN 186311) 113.08 µM trolox·g^−1^ Churumbinho (186305) 70.05 µM trolox·g^−1^	Eight pepper genotypes (*Capsicum sp*., *Capsicum annun* L., C. *chinense* Jacq, and C. *baccatum* L. var. *umbilicatum*)	[7]
	Carotenoids, vitamin C, and phenolic compounds	Bell pepper (*Capsicum annuum* L.). Cultivar/rootstock combinations: Jeanette/Terrano (yellow), Sweet/Robusto (green), Fascinato/Robusto (red), Orangela/Terrano (orange), and Fascinato/Terrano (red)	Fascinato/Robusto (79.65% inhibition) Orangela/Terrano (76.0% inhibition) Fascinato/Terrano (73.5% inhibition) Sweet/Robusto (64.90% inhibition) Jeanette/Terrano (64.90% inhibition)	Commercial varieties of bell pepper were used as scions and grafted from either Terrano or Robusto rootstock	[68]
	Phenolic compounds (flavonoids) and capsaicinoids	Twenty chili cultivars belong to *Capsicum annuum*, *Capsicum baccatum*, *Capsicum chacoense*, and *Capsicum chinense*. Bell, orange Habanero, Cayenne, red Habanero, Malagueta, and Dedo de moça peppers	DPPH assay: Effix (C. *annuum*) IC_50_ = 3.9 µg·mL^−1^ in fresh pepper; Loco (C. *annuum*) IC_50_ = 28.1 µg·mL^−1^ in boiled pepper; Acrata (C. *annuum*) IC_50_ = 5.0 µg·mL^−1^ in frozen pepper ABTS: Nobile and Acrata. (C. *annuum*) IC50 = 26.5 and 27.3 µg·mL^−1^ in frozen pepper	Fresh, boiled and frozen chili peppers cultivars belonging to four *Capsicum* species	[69]
	Phenolic compounds (flavonoids) and capsaicinoids	Red chili pepper seeds (*Capsicum frutescens* L.)	Seeds extracts with n-hexane (DPPH = 28% at 1000 μg·mL^−1^ and seeds extracts with chloroform (DPPH = 29% at 1000 μg ·mL^−1^)	Seeds from *Capsicum frutescens* L.	[65]
*Anti-hypertensive activity*	Phenolic compounds (flavonoids)	Medicinal plants, herbs, and species commonly used in Latin America: Arbol, Ancho, and Rocoto chili peppers (*Capsicum*)	2.5 mg of dried sample (% ACE-inhibition): chile de arbol 45%, chile ancho 68%, Japanese chili 68%, paprika pepper 92 %, yellow pepper 48%, red pepper 84%, and rocoto 70%	In vitro potential against enzymes for hypertension of several chili peppers (*Capsicum*).	[65]
	Phenolic compounds (flavonoids) and capsaicin	Various tissues of red chili pepper (*Capsicum annuum* L.)	Red pepper pericarp A (97% at 5 mg·mL^−1^ of extract) placenta A (64% at 5 mg·mL^−1^ of extract) stalk A (14% at 5 mg·mL^−1^ of extract); red pepper pericarp B (90% at 5 mg·mL^−1^ of extract) placenta B (54% at 5 mg·mL^−1^ of extract) stalk B (16% at 5 mg·mL^−1^ of extract)	In vitro inhibitory potential ACE-inhibition against hypertension of red pepper (*Capsicum annuum* L.)	[3]
*Anti-hyperglycemic activity*	Phenolic compounds (flavonoids), carotenoids, and capsaicinoids	Habanero chili pepper (*Capsicum chinense* Jacq. cv. Habanero)	Total extract: α-amylase immature IC_50_ = 229 μg·mL^−1^, mature IC_50_ = 131 μg·mL^−1^; α-glucosidase immature IC_50_ = 150 μg·mL^−1^, mature IC_50_ = 265 μg·mL^−1^	Fruits of *Capsicum chinense* Jacq. cv Habanero harvested at the same time but at two successive maturity stages	[17]
	Phenolic compounds (flavonoids)	Spices, chili peppers, medicinal plants, and herbs were evaluated. The extracts from chili peppers analyzed were: Arbol, Ancho, Yellow, Japanese, Red, Paprika, and Rocoto chili pepper (*Capsicum*)	2.5 mg of dried sample (% Glucosidase inhibitory activity): chile de arbol 20%, chile ancho 23%, Japanese chili 38%, paprika pepper 30%, yellow pepper 40%, red pepper 45% and rocoto 46%	In vitro potential against enzymes for hyperglycemia of several chili peppers (*Capsicum*).	[65]
	Phenolic compounds (flavonoids), carotenoids, and capsaicinoids	The varieties of chili pepper assessed in this study were: Fiesta, Acuminatum, Orange Thai, and Golden Cayenne (*Capsicum annuum* L.)	Total extract (α-glucosidase activity): Fiesta immature (IC_50_ = 109.2 µg·mL^−1^), Fiesta mature (>1000), Orange Thai immature (IC_50_ = 102.5 µg·mL^−1^), Orange Thai (IC_50_ = 166.5 µg·mL^−1^), Acuminatum immature (>1000), Acuminatum mature ((IC_50_ = 71.5 µg·mL^−1^), Cayenne Golden immature (IC_50_ = 81.1 µg·mL^−1^), Cayenne Golden mature (IC_50_ = 63.6 µg·mL^−1^) and Acarbose (IC_50_ = 35.5 µg·mL^−1^)	Four *Capsicum annuum* L. cultivars were studied at two stages of fruit ripening (immature and mature)	[70]
	Phenolic compounds (flavonoids) and capsaicinoids	Various tissues of red chili pepper (*Capsicum annuum* L.)	Red pepper pericarp A (58% at 5 mg·mL^−1^ of extract) placenta A (38% at 5 mg·mL^−1^ of extract) stalk A (40% at 5 mg·mL^−1^ of extract); red pepper pericarp B (52% at 5 mg·mL^−1^ of extract) placenta B (32% at 5 mg·mL^−1^ of extract) stalk B (60% at 5 mg·mL^−1^ of extract)	In vitro inhibitory potential of α-glucosidase against hyperglycemia of red pepper (*Capsicum annuum* L.)	[3]
Metal-chelating activity	Phenolic compounds (flavonoids) and capsaicinoids	Habanero chili pepper (*Capsicum chinense* Jacq. cv.)	71% at 42 days after fruit set	Habanero chili pepper (*Capsicum chinense* Jacq. cv.) examined during nine maturity stages (at 7-day intervals from fruit set)	[19]
Antitumoral activity	L-asparaginase	Green chili pepper (*Capsicum annuum* L.)	Maximum cell growth inhibition was observed in the Human Oral Squamous Carcinoma cell line (IC_50_ 360 µg·mL^−1^), while the least activity was found in Human Lung Carcinoma lines (IC_50_ 535 µg·mL^−1^) and moderate activity in Human, Cervix cell lines (IC_50_ 410 µg·mL^−1^)	Antiproliferative activity of L-asparaginase against three human cancerous cell lines	[20]
	Capsaicinoids and carotenoids	Algerian chili pepper (*Capsicum annuum* L.)	The antitumor activity of capsaicinoids at a concentration of 200 μg·mL^−1^ was 52%, and carotenoids reached 90% activity at the same concentration	Antitumor potential of capsaicinoids and carotenoids against cancerous (U937) and healthy (PBMC) cell lines	[9]

**Table 3 molecules-28-04239-t003:** Commercial products and cosmetics with ingredients derived from *Capsicum*.

Product	Brand	Type of Product	Beneficial Effect	References
Dermatologic cream	AG Cosmética natural™	Cosmetic	Antihyperglycemic and anti-inflammatory	[150]
Paprika and chili balm massage gel	Dr. C. Tuna™	Cosmetic	Anti-inflammatory and antioxidant	[151]
Chili pepper (*Capsicum*) oil-based extract	Flowertales™	Cosmetic	Anti-inflammatory and antioxidant	[152]
Capsicum body soap	S-SKIN Naturals™	Cosmetic	Antioxidant	[153]
Facemask Kawaii red pepper	Gipsy vibes™	Cosmetic	Antioxidant	[154]
Red pepper oil (*Capsicum* oil)	Mani chemicals™	Cosmetic	Anti-inflammatory and antioxidant	[155]
Capsicum (oleoresine)	Venkatramna™	Cosmetic	Anti-inflammatory and antioxidant	[156]
Cream with CBD and capsaicin	Oliver’s Harvest™	Cosmetic	Antihyperglycemic	[157]
Plant thermoactive anti-cellulite oil with hot pepper extract	Cosmetic plant™	Cosmetic	Anti-inflammatory and antioxidant	[158]
Capsaicin sauce	Jayone™	Foods	Antioxidant	[159]
Red pepper paste	Galil™	Foods	Antioxidant	[160]

## Data Availability

Not applicable.

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
