# Peer review of "The Genus Capsicum: A Review of Bioactive Properties of Its Polyphenolic and Capsaicinoid Composition"

_molecules, 2023, doi:10.3390/molecules28104239_

Round 1
Reviewer 1 Report
Article with a current and interesting theme. However incomplete. I suggest that for the publication the authors insert the application of peppers in food, beverages, and cosmetics, relating their antioxidant capacity with the previously mentioned beneficial effects. I believe a table with the applications and a brief critical discussion about them complete the article for publication.
Author Response
"Please see the attachment."

Reviewer 2 Report
Although the review article is of high interst and beneficial for the readers of the Journal it needs to be revised taking into consideration the following comments:
1. Large part of the article ( paricularly the 1 and 2 sub-titles) was not prepared as a review, it is rather a Chapter ina book not review of the recent achievements and findings in the field.
2. I noticed a confusion in the main target of the review. Is it about chili peppers or the crops of the genus Capsicum, which include both pungent and non-pungent peppers. It would bo mre correct if it deals with the chili peppers only. The title should be changed accordingly.
3. The English style used in many paragrapgs is not for a scientific review, sometime I feel that it is a general article for News letter not for scientific journal Like Molecules. I try to mention some of the weak or broken phrases:
Line 18. The sentence should read " ..of the saLonaceae famikly, which includes eggplant, tomatoes, potato, and tobacco."
Abstract: The last part should be corrected to " This review describes the data from in-vivo and in-vitro evaluation of the bioactive properties attributed to poly phenols and capsaicinoids of the different chili pepper products. Such data are helpful in formulating the functional foods or food ingredients.
Line 36 and 41. the sentence is mentioned aerlier (repeated)
Line 44. The same as suggested for line 18.
Line 47. from "they show better growth.." to be corrected to: However, tropical and sub-tropical environment is the optimal for the plant growth.
Line53. What is (Ruiz and Pav)??
Line 68. The sentence should read" The products of chili peppers contain nutritionally important metabolites"
Line 71. the sentence from dectrease tghe concentration of..... is imprecise expression. The antioxidant activity is due to oxygen quanching and free radical scavenging capacity of the compounds.
Line 73. The sentence is weak and the phrase is broken.
Line 74. The sentence should be corrected to: Chili pepper is usually consumed fresh and processed in different forms like spice powder, paste etc.
Line 75-76. The two sentences should be mentioned aerlier not here with a part dealing with the nutrional importance.
Line 101.The sentence is not acceptable (not correct)
Table 2. The content of the table is consusing, of no interest in this form. It cannot be followed .
Line 128. which compounds are involved (ROS or Phenolics)? to be clarified
Line 140. In steade of can be found, I suggest "they occure free or esterified as glucoside.
Line 159. I miss the data on luteoline , which has been reported to be one of the dominant poly phenols in bell peppers or chilis.
Line 166-167. The sentence obout anthiocyanins to be rewritten as : anthocyanins (delphinidin glucosides) have been reported as compounds responsible for the colour of such varieties.
Line 170-200. This part is prepared in a way suitable for Chapter in book not review. The same hold true for carotenoids. The is a marked generality of the topcs.
Line 345. Something is missing in the sentence
Part 3. (3.1., 3.2., 3.3., to 3.6.) is well prepared. However the authors used the passivoice verbs, which is usually used in the desciption of the own research article. The correct is the present perfect when the authors use the data from the literatures ( has or have been stated, reported, found, etc.)
I included it in the previous part of my review
Author Response
"Please see the attachment."

Reviewer 3 Report
A summary of research on the antioxidant, antimicrobial, anti-inflammatory, antihypertensive, antihyperglycemic, metal-chelating, and antitumor effects of polyphenols and capsaicinoids in chili peppers is presented in this article.
While the topic of this work might be interesting to the scientific community working on the effects of bioactive plant compounds such as those derived from the genus Capsicum, the work itself needs improvement.
If accepted, the manuscript will require some additional technical editing. I have pointed out some, but not all, areas that require technical editing in my comments.
Comments:
1. To improve the value of table 2 and the readability of the article, I would strongly suggest adding two more columns to this table showing the models/cell lines and doses/concentrations studied.
2. Wouldn't it be better to list specific ingredients instead of "phenolic compounds" in the table 2?
3. In my opinion the descriptions in subsection 2.2. "Bioactive Compounds Found in Chili Peppers" of flavonoids, capsaicinoids and capsinoids are too extensive. They should be shortened.
4. In the subsections on activities 3.1-3.7 only descriptions of the collected literature data are presented. The results would be enhanced by a short summary following each subsection. Are these all the activities of the Capsicum genus described in the literature? In addition, in section 3.7 (anti-cancer potential) there is only one study conducted on U937 cancer and healthy (PBMC) cells. Meanwhile, for example, in the PubMed database, you can find many other works on the anticancer activity of the Capsicum genus. Please, check.
5. Moreover, are there any human studies on this topic in the literature? If there are any, please include them.
6. It is definitely necessary to include the authors' own opinions or insights in the text.
7. Conclusion section is too general.
8. Many sentences need to be revised and clarified. The authors state, for example, in line 476: “Plant extracts are now seeking natural alternatives to control hyperglycemia as an alternative to existing drug-based therapies. It is impossible for polyphenols to look for alternatives...and this is only one example.
1 The manuscript is filled with repetitions that make it difficult to read. There are also multiple grammatical issues throughout the manuscript. Authors are advised to proofread the whole manuscript to overcome these mistakes and make it easier for the reader.
Author Response
"Please see the attachment."

Round 2
Reviewer 2 Report
As I mentioned in the comments on the quality of English language
There is a miss use to the present perfect style . It should be used as follows:
After the authors names the verb to b in the past form, e.g., Britton et al, (2010) found ( not have been found)
Have (has) been found should be after the targets e.g., Capsaicinoids have been reported to be... Lycopene has been evaluated...
Reviewer 3 Report
Thank you for responding to my comments. The manuscript is ready for publication.